# Comparative electron microscopy analysis of internal limiting membrane and epiretinal membrane ultrastructure from vitrectomy surgery: A study protocol

**Thananop Pothikamjorn**[1], **Thanapong Somkijrungroj**[2,3], **Marisa Prasanpanich**[4], **Nuntachai Surawatsatien**[2,3], **Wasee Tulvatana**[3]*

1 Faculty of Medicine, Chulalongkorn University and King Chulalongkorn Memorial Hospital, Thai Red Cross Society, Bangkok, Thailand, 2 Center of Excellence in Retina, Faculty of Medicine, Chulalongkorn University and King Chulalongkorn Memorial Hospital, Thai Red Cross Society, Bangkok, Thailand, 3 Department of Ophthalmology, Faculty of Medicine, Chulalongkorn University and King Chulalongkorn Memorial Hospital, Thai Red Cross Society, Bangkok, Thailand, 4 Department of Pathology, Faculty of Medicine, Chulalongkorn University and King Chulalongkorn Memorial Hospital, Thai Red Cross Society, Bangkok, Thailand

* waseetulvatana@chula.md

## Abstract

The epiretinal membrane (ERM) is a fibrocellular layer that forms on the inner surface of the retina, often leading to visual impairment and significantly impacting visual function. Understanding the pathophysiology of ERM formation is crucial for advancing ophthalmic care and improving patient outcomes. This research aims to investigate the pathophysiology of ERM by comparing the internal limiting membrane (ILM) pathology in patients with and without ERM. The study involves a comprehensive protocol of ILM and ERM specimens collection during pars plana vitrectomy and membrane peeling procedures. Specimens are assessed using both light microscopy and transmission electron microscopy to evaluate morphological and ultrastructural changes. The study employs a standardized protocol for specimen collection and analysis, focusing on identifying differences in cell counts, extracellular matrix components, and ultrastructural alterations in the ILM. We also investigate the correlation between pathological findings and clinical biomarkers, including fundus photography, optical coherence tomography (OCT), optical coherence tomography angiography (OCT-A), and baseline characteristics such as patient demographics and underlying diseases. These clinical assessments provide a comprehensive understanding of the ocular environment and its relationship to ERM formation. By examining both the ILM and ERM in patients with and without ERM, the study aims to identify distinct pathological features associated with ERM development. We also aimed to elucidate whether changes in the ILM, such as cellular proliferation and extracellular matrix remodeling, are significant contributors to ERM formation. Additionally, the study will explores how these pathological changes correlate with clinical features and

**Data availability statement:** The study protocol is accessible at https://www.protocols.io/ (dx.doi.org/10.17504/protocols.io.81wgb-w613gpk/v1). All individual participant data collected during the study, after de-identification, are available at https://figshare.com/ (figshare.com/s/8f57d71f954b909d6d08?-file=57227747).

**Funding:** This study was funded by the Ratchadapiseksomphot Fund, Faculty of Medicine, Chulalongkorn University (Grant number MDCU GA67/038) (WT), the 90th Anniversary of Chulalongkorn University Ratchadapiseksomphot Research Funds (Grant number GCUGR1125671149M) (WT), and the grants for research of the Center of Excellence in Retina, Faculty of Medicine, Chulalongkorn University and King Chulalongkorn Memorial Hospital, Thai Red Cross Society (Grant number 25010119) (TP, TS). The funders have no role in study design, data collection and analysis, decision to publish, or preparation of the manuscript.

**Competing interests:** The authors have declared that no competing interests exist.

biomarkers, offering insights into potential mechanisms driving ERM pathogenesis. Establishing these correlations would support the hypothesis that ILM changes contribute to ERM development, while the absence of significant differences may suggest alternative pathways. Ultimately, this research aims to enhance our understanding of ERM pathophysiology, paving the way for improved prognosis and therapeutic strategies.

## Introduction

The epiretinal membrane (ERM), also known as the epimacular membrane (EMM), is a common yet often overlooked retinal condition that can impact vision [1]. This thin, fibrocellular membrane forms on the retina's surface, leading to symptoms such as metamorphopsia or visual impairment. If left untreated, ERM can progressively lead to substantial vision loss[2]. The global prevalence of ERM varies widely, ranging from 1.9% to 28.9%, lower in Asians such as Chinese [3,4] or Japanese [5,6] than Caucasians [1,7], depending on different factors within the population characteristics [8].

ERM is categorized into primary (idiopathic) and secondary forms. Secondary ERM is commonly associated with other ocular diseases such as retinal break, retinal detachment (RD), diabetic retinopathy, retinal vascular disease, ocular inflammation, and vascular occlusion [9, 10]. Demographic factors such as sex, age, and ethnicity also influence the likelihood of developing ERM, with incidence increasing with age and being associated with systemic conditions like diabetes mellitus (DM), hyperlipidemia, and hypercholesterolemia [1, 7, 8, 11]. Despite the diverse causes, many cases remain idiopathic with unclear pathophysiology of membrane formation.

The primary treatment for ERM is a surgical procedure called pars plana vitrectomy (PPV), where the surgeon removes the ERM and some practices also peel the internal limiting membrane (ILM) to improve outcomes such as reduce ERM recurrence [12,13]. In participants with macular hole, ILM peeling is used prophylactically to prevent secondary ERM [14,15]. This suggests that the pathology of the ILM may be associated with the development of ERM, particularly in diseases with ILM defects, such as retinal breaks or macular holes. The ILM may play a role in cellular migration, contributing to ERM formation [9]. Comparing ILM pathology between participants without ERM and those with ERM, especially between idiopathic and secondary ERM, may reveal pathological features related to idiopathic ERM formation. This comparison is crucial for understanding the relationship between ILM and ERM and addressing the gap in pathophysiology of membrane formation.

Previous research has identified various pathological features within the ERM and ILM, including the presence of retinal pigment epithelium (RPE), fibrous astrocytes, fibrocytes, myofibrocytes, and myofibroblasts [16,17]. RPE cells, typically found in the retina's outer layers, dominate ERM pathology, suggested their migration to the membrane [16]. Other changes include cellular fragments, vacuolization, fibrillary structures, and hyaloid attachment, alongside alterations in cellular structures like newly formed collagen or organelles such as nucleus, nucleolus, rough endoplasmic reticulum, and mitochondria [16–20].

In addition to comparing ILM without ERM and ILM with ERM pathologies, analyzing ILM pathologies between participants with idiopathic and secondary ERM may reveal histopathologic differences. Previous studies have documented these changes using both light microscope (LM) and electron microscopy (EM) and other biomarkers such as OCT, fundus photography, and OCT-A, yet a comprehensive comparison between pathological and normal ILM remains unexplored [17,18]. Our study aimed to report pathological results of ILM and ERM comparing ILM without ERM and ILM with ERM and their correlation with clinical biomarkers and patient characteristics.

Furthermore, there is a lack of literature on standardized specimen collection and processing, as well as pathological comparisons of ILM in patients with ERM versus those with other conditions requiring ILM peeling. This protocol aims to establish a standardized method for specimen harvesting, coding, processing, evaluating, and reporting ILM and ERM pathological results using LM and EM and correlate them with clinical biomarkers and patient characteristics.

The primary objective is to study the electron microscope morphology and pathological changes of cells and extracellular matrix findings in ILM without ERM and ILM with ERM. The secondary objectives are to develop a standardized method for specimen collection and processing for electron microscopy examination, to compare the electron microscope histopathological findings of ILM, primary ERM, and secondary ERM specimens, and to analyze participants' baseline characteristics associated with the presence and absence of ERM and correlate these with the electron microscope histopathological findings of cells and extracellular matrix in the ILM. Detailed data on hypothesis and plan are demonstrated in Table 1.

## Materials and methods

### Ethics information

This protocol and informed consent forms have been approved by the Institutional Review Board of the Faculty of Medicine, Chulalongkorn University (COA No. 0948/2023). Participants must provide written informed consent prior to any study procedures. We adhered to the principles outlined in the Declaration of Helsinki.

### Design and setting

This study is a pathologist-masked and ophthalmologist-masked cross-sectional analytical study that aims to compare the LM and EM pathologic results of the ILM with ERM from participants undergoing ERM peeling with ILM peeling and ILM without ERM in participants with ILM peeling. All participants will be consecutively recruited from the outpatient ophthalmology clinic of two retina surgeons (TS and NS) at the Department of Ophthalmology, King Chulalongkorn Memorial Hospital. Study participants will then be allocated to each group based on clinical diagnosis, imaging, and surgical procedures performed, which must include membrane peeling and may involve additional procedures such as phacoemulsification, silicone oil removal, or implantation, or intravitreal injection. All surgeries will be performed by two highly experienced retina surgeons (TS and NS), with participants recruited from their respective outpatient clinics. Participants will undergo routine post-operative evaluations over a 3-month period to assess surgical outcomes. The study will objectively measure the electron microscope morphology and pathological changes of cells and extracellular matrix findings in ILM without ERM and ILM with ERM, compare them with subgroup analysis of primary ERM and secondary ERM, and analyze the correlation between the pathologic results and participants' baseline characteristics, clinical biomarkers, and surgical outcomes.

### Sample size

Given the limited data from available studies on the prevalence of ILM peeling performed in a single center, we estimated our sample size based on previous operations conducted three months before the study. We then extrapolated this to a one-year specimen collection period to estimate our sample size, although the actual number may vary depending on the surgical capacity and availability of both surgeons. We anticipate collecting approximately 60–80 specimens of ILM with ERM and 10–20 specimens of ILM without ERM.

**Table 1. Detailed data of hypothesis and plan.**

| Question | Hypothesis (if applicable) | Sampling plan (e.g., power analysis) | Statistical Analysis Plan | Interpretation given to different outcomes |
|---|---|---|---|---|
| What are the morphologic and pathologic changes of cells and extracellular matrix from the light and electron microscopic findings of ILM in patients with ERM and patients who need ILM peeling for secondary ERM prophylaxis? | There are significant pathological changes of ILM between ILM in patients with ERM and patients who need ILM peeling for secondary ERM prophylaxis. The null hypothesis for these tests is that there is no difference in the proportion of specific histopathological findings between the groups. | We estimated our sample size based on previous operations conducted three months before the study. We anticipate collecting approximately 60–80 specimens of ILM with ERM and 10–20 specimens of ILM without ERM. | Descriptive statistics will be used to characterize the electron microscope findings of cellular and extracellular matrix morphology and pathological changes in ILM samples, both with and without ERM. This will include calculating means, standard deviations, medians, interquartile ranges, and frequency distributions, as appropriate. A two-proportion Z-test will be conducted to compare the histopathological findings observed under electron microscopy between ILM samples with ERM and those without ERM. | Morphology and pathological changes of cells and extracellular matrix of ILM may relate to the cause of ERM. |
| What is the dominance cell types and their pathological changes of cells and extracellular matrix of ILM and ERM from electron microscopic findings between ILM peeled with ERM of patients with Idiopathic ERM and Secondary ERM? | There are significant numbers of dominance cell types and pathological changes of ILM and ERM between patients with Idiopathic ERM and Secondary ERM. | As primary outcome. | Descriptive statistics will be used to characterize the electron microscope findings of cellular and extracellular matrix morphology and pathological changes in ILM and ERM samples, including both primary and secondary ERM. This analysis will include calculating means, standard deviations, medians, interquartile ranges, and frequency distributions, as appropriate. A two-proportion Z-test will be conducted to compare the histopathological findings observed under electron microscopy between primary ERM and secondary ERM. | The dominance cell types and their pathological changes of cells and extracellular matrix of ILM and ERM from electron microscopic findings may differentiate Idiopathic ERM and Secondary ERM. |
| What is the correlation between the ILM and ERM pathological results and participants' baseline characteristics and clinical biomarkers. | Participants' baseline characteristics and clinical biomarkers has a significant correlation to detect the changes of ILM in patients having ERM. | As primary outcome. | We will first assess the distribution of our count data to determine if it fits a Poisson distribution. If the data are suitable, we will perform Poisson regression; otherwise, alternative count models, such as negative binomial regression, will be considered. We will also conduct a multivariate analysis to account for potential confounding factors. The regression models will include variables such as demographic characteristics (age, sex, ethnicity) and clinical biomarkers (OCT, fundus photography, OCT-A results). | Participants' baseline characteristics and clinical biomarkers may distribute a significant correlation to detect the changes of ILM and ERM pathological results in patients having ERM. |

## Participants' characteristics

Participants undergoing PPV for ILM peeling and/or ERM peeling, who meet the eligibility criteria, will be invited to participate in the study. Screening for eligibility will be conducted independently by retina specialists (TS and NS), who are responsible for specimen harvesting. Participant enrollment and project activities are scheduled to commence in August 2023 and conclude in October 2024. Study inclusion and exclusion criteria are detailed in Table 2.

## Study methodology

### Masking

Pre-coded 1.8 mL cryovial tubes will be pre-stored at the operating theatre. Upon harvesting, samples will be immediately stored in each designated tube. The retina surgeon will report surgical findings using the pre-assigned tube code, ensuring masking of the EM laboratory specialist and pathologist (MP) for evaluation using both LM and EM.

**Table 2. Inclusion and exclusion criteria of the study.**

| ILM without ERM | | ILM with ERM | |
|---|---|---|---|
| Inclusion criteria | Exclusion criteria | Inclusion criteria | Exclusion criteria |
| 1. Participants who require ILM peeling, such as for macular hole closure or ERM prophylaxis in those with RD.<br>2. Have been investigated by OCT preoperatively<br>3. Acquired both the LM and EM pathological results. | 1. Diseases or surgical conditions in which specimen harvesting would complicate the procedure or adversely affect the participant's outcome, such as highly complex cases where tissue collection may hinder the surgeon's ability to perform a safe surgery.<br>2. The previous eye has been recruited to the study | 1. Participants with idiopathic ERM or secondary ERM surgery with ILM peeled at King Chulalongkorn Memorial Hospital<br>2. Have been investigated by OCT preoperatively<br>3. Acquired both the LM and EM pathological results. | 1. Diseases or surgical conditions in which specimen harvesting would complicate the procedure or adversely affect the participant's outcome, such as highly complex cases where tissue collection may hinder the surgeon's ability to perform a safe surgery.<br>2. The previous eye has been recruited to the study |

EM: Electron microscope; ILM: Internal limiting membrane; LM: Light microscope; OCT: Optical coherence tomography; RD: Retinal detachment.

OCT, fundus photography, and OCT-A reports will be conducted by a retina specialist (TS). The images will be assigned a second set of codes, which will later be matched with the first set of codes assigned to the specimens for result correlation.

## Specimen harvesting

Each participant will undergo their surgical procedures as usual, which will be either a 25G or 27G PPV with ILM and/or ERM peeling. During the membrane peeling process, the surgeon will stain the membrane using brilliant blue G until it is visualized for peeling. Subsequently, the surgeon will perform continuous curvilinear membrane peeling using forceps, targeting either the ILM alone or both the ILM and ERM, depending on intraoperative findings. In cases where the ILM and ERM are present together, we aim to collect both membranes for histopathological examination. If the ILM and ERM are separated, multiple peels may be required, and all tissue will be collected into the same specimen tube. In cases where the ILM and ERM are inseparable, the membranes may be removed in one or more passes at the surgeon's discretion to obtain the maximum amount of tissue for processing. All specimens will be extracted from the intraocular space via the vitrectomy port using forceps. Afterward, the specimens will be placed on sterile filter paper shown in Fig 1, folded, and fixed in the tube. The method of specimen harvesting is demonstrated in S1 Video.

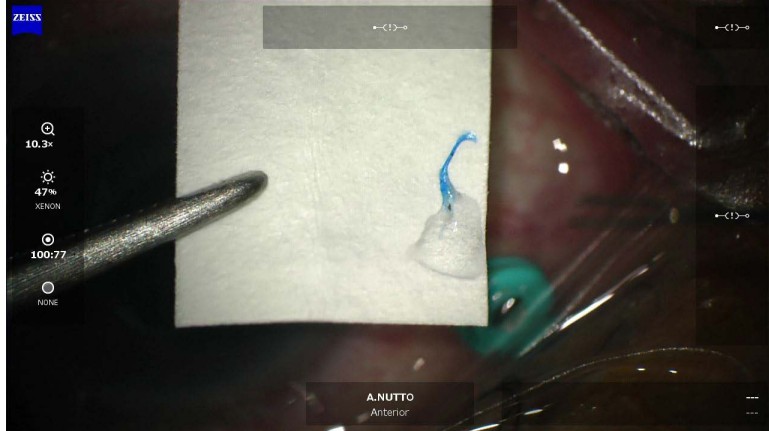

**Fig 1. Specimen harvested placed on sterile filter paper.**

## Specimen coding and delivery

One-hundred and twenty pre-coded 1.8 mL cryovial tubes, each prefilled with 3% glutaraldehyde in 0.1M phosphate buffer at pH 7.3, will be stored in a temperature-controlled refrigerator at 4°C in the operating theatre. During each specimen harvest, the circulating nurse of each surgery will randomly select one tube. The retina specialist will then harvest the specimen from intraocular, immediately place it on sterile filter paper, fold it, and fix it in the tube. After fixation, the tube with specimens will be stored at another location in the temperature-controlled refrigerator at 4°C for 2–14 days in the operating theatre before being delivered to the EM laboratory.

Upon arrival at the laboratory, each tube will undergo a phosphate buffer washout to replace the 3% glutaraldehyde with 0.1M phosphate buffer at pH 7.3. Subsequently, the specimen will be fixed in 4% osmium dioxide, causing it to change color to a dark brown hue, and then embedded in gelatin, with the filter paper removed. If the specimen is not visible at this stage, the process will be halted, and it will be marked as specimen loss during delivery. If the specimen is visible, it will undergo dehydration in graded ethanol, followed by embedding in resin and processing using an ultramicrotome cutter to obtain the specimen layer for LM examination. The pathologist will then report the presence of the specimen using LM. If the specimen is absent at this stage, the EM examination will be discontinued, and it will be marked as specimen loss during processing. If the specimen is present, it will be examined using the JEOL-JEM 1400 plus transmission electron microscope (TEM). The specimen processing and microscopic examination methods was adapted from previous studies [16-18,21,22].

## Light microscope examination

Pathologists will assess each specimen and determine its presence or absence. If present, the specimen's colorization will be described as either adequate or inadequate. Further examination will involve cell counting, conducted at a high-power field (400x), with up to 10 fields examined (1–10 depending on specimen size). The cell count will be calculated as cells per 1 mm$^2$. Additional descriptions will be provided for any significant findings observed during the examination.

## Transmitted electron microscope examination

In the JEOL-JEM 1400 plus TEM, the scan results will be marked by a grid. We will select the grid with the highest specimen-to-grid ratio. Ultrastructure examination for TEM is described in Table 3.

**Table 3. Detailed data of ultrastructure examination at each zoom power.**

| Structure | Zoom power | Ultrastructure examination |
|---|---|---|
| Internal limiting membrane | 400x | 1. Whole grid examination – marking maximum of 10 boxes for ultrastructure examination (depends on specimen sizes). Zooming in can be done to confirm each box has the ILM within the grid. |
| | 8000x | 1. Vacuolization examination – examining for vacuolization of all selected boxes. Zooming in can be done to assess the appearance of the vacuoles.<br>2. ILM surface description – both retinal side and vitreous side |
| | 20000x | 1. ILM appearance description – such as degree of looseness. |
| Epiretinal membrane | 400x | 1. Whole grid examination – marking the presence of the ERM and select the grid with highest cellularity for the description of the cell types. The characteristics of each cell type were as mentioned by Smiddy et al. [16], Gandorfer et al. [17], Beyazyildiz et al. [18], and Frisina et al. [22] |
| | 20000x | 1. Content description – such as native collagen fibers and newly formed collagen. The appearance of the collagen fibers was as mentioned by Smiddy et al. [16], Gandorfer et al. [17], Beyazyildiz et al. [18], Kritzenberger et al. [19], and Regoli et al. [20] |

We will mark the interesting point, note in description, and capture them for future selection in manuscript publishing.

## Data management

After pathologic results are reported, each specimen will be coded with other sets of codes, later matched with the first code to match variables collected in the study. The variables collected are demonstrated in Table 4.

## Statistical analyses plan

From Table 1, all analyses will be performed using the statistical software STATA SE 18 (StataCorp LLC, College Station, Texas, USA). A significance level of 0.05 will be used for hypothesis testing. This statistical analysis plan is designed to ensure a comprehensive evaluation of the morphologic and pathologic changes in ILM specimens and their association with clinical and surgical variables. Loss of tissue during transfer or specimen handling will be reported in percentages. The missing patient characteristics will be imputed using multiple imputations. Sensitivity analysis may be conducted by excluding specimens with distinctly fewer boxes reported for pathological results in both LM and TEM.

The classification of ILM with or without ERM for analysis will be based on the current gold standard of diagnosis, OCT. Additionally, the surgical diagnosis at the time of specimen collection will be recorded, as both ILM and ERM may occasionally be identified intraoperatively using brilliant blue G staining, particularly in cases where preoperative OCT is of poor quality. Such scenarios may arise in participants with vitreous hemorrhage, dense cataract, or severe RD. These cases will be categorized as "cannot evaluate" in our analysis and will be appropriately stratified.

## Status and timeline

The recruitment process started on August 21st, 2023, was expected to end on October 11th, 2024. The study was anticipated to be completed by August 2025.

**Table 4. Detailed data of variables collected.**

| Variables | Factors collected |
|---|---|
| **Participants characteristics** | 1. Age, collecting birthdate and surgery date<br>2. Sex<br>3. Ethnicity |
| **Diagnosis** | 1. Diagnosis of disease which requires membrane peeling<br>2. Underlying disease, mainly hypertension, dyslipidemia, and diabetes mellitus (staging as no diabetic retinopathy, nonproliferative diabetic retinopathy, and proliferative diabetic retinopathy).<br>3. Cause of secondary ERM, including DM with NPDR or PDR, RD, retinal vascular occlusive diseases, uveitis, retinal breaks seen during surgery or from fundus photo and vitreous status such as post-vitrectomized and silicone oil. |
| **Clinical biomarkers** | 1. Eye side<br>2. Fundus photo<br> 2.1. Overall findings of fundus photo, such as retinal break or detachment, vitreous hemorrhage, high myopic fundus<br> 2.2 Age-related macular degeneration (AMD) classification adopted from Ferris et al. [23]<br> 2.3. Pigmentation classification adopted from Mitchell et al. [1]<br>3. OCT – biomarkers including<br> 3.1. Central subfoveal thickness (CST)<br> 3.2. Vitreomacular adhesion (VMA) or traction (VMT) syndrome as focal or broad with a cut point width of >1500 µm adopted from Johnson et al. [24] and related markers including intraretinal cyst, lamellar or full thickness macular hole (FTMH)<br> 3.3. Perifoveal vitreous detachment (PVD) staging adapted from Johnson et al. [25]<br> 3.4. ERM staging adapted from Govetto et al.[26] and related markers such as disorganization of retinal inner layers (DRIL)<br> 3.5. Other markers such as retinoschisis or loss of subfoveal ellipsoidal band<br>4. OCT-A – increased foveal avascular zone (FAZ) |
| **Surgical outcomes** | 1. Visual acuity at 3 months post-operative<br>2. Complications |

## Discussion

Our study aims to enhance the pathophysiological understanding of the ILM and its relationship with ERM development. We seek to identify the EM ultrastructural pathological changes associated with primary and secondary ERM, as determined by clinical and pathological diagnoses. By collecting comprehensive data on participants' characteristics, clinical biomarkers, and surgical outcomes, we aim to elucidate the relationship between these factors and their pathological results. This cross-sectional database, enriched with diverse participant characteristics and clinical biomarkers, will inform future pre-emptive and early intervention studies related to ERM.

This protocol is the first to study intraocular specimens from vitreoretinal surgery using both LM and TEM. However, the study has some limitations. This type of research is inherently restricted to a cross-sectional design due to the removal of the ILM, precluding the study of future ILM regeneration in participants. Additionally, the early start of enrolment following the COVID-19 pandemic slowed recruitment and pathological reporting, resulting in a limited sample size based on the number of surgeries performed within a year. The lack of data on the prevalence of ILM removal surgeries for other diseases necessitated using our site's timeframe for sample size calculation. Furthermore, potential biases may arise from our unique specimen collection and reporting methods, which, while novel and replicable, may have overlooked relevant pathological findings and participant characteristics pertinent to ERM presence.

We recognize the following limitations in our study design and methodology. Given the cross-sectional study design, our ability to draw causal inferences regarding ERM development and progression is limited. Additionally, this design does not allow for the assessment of ILM regeneration, ERM recurrence, or long-term visual outcomes. A longitudinal study would be required to correlate ultrastructural findings with disease progression and clinical outcomes over time. Lastly, we acknowledge that our study population was recruited from a single tertiary care center, which may introduce selection bias due to institutional surgical preferences or referral patterns. This limitation may affect the generalizability of our findings to broader or more diverse populations and surgical settings.

## Supporting information

**S1 Table. STROBE Statement.** Checklist of items that should be included in reports of cross-sectional studies available in supplementary material.
(DOC)

**S1 Video. Method of specimen harvesting.**
(MP4)

## Acknowledgments

We would like to extend our gratitude to Sirinapa Srikam and Wilawan Ji-au, EM laboratory specialists of the Department of Pathology, Faculty of Medicine, Chulalongkorn University and King Chulalongkorn Memorial Hospital, Thai Red Cross Society, Bangkok, Thailand, as well as Dr.Wajamon Supawatjariyakul of the uveitis unit and Dr.Chanida SareeKhome, Dr.Nathapon Treewipanon, Dr.Patthicha Pinyosawadsakul, and Dr.Pimpisa Vudhichaiphun of the retina unit, Center of Excellence in Retina, Faculty of Medicine, Chulalongkorn University and King Chulalongkorn Memorial Hospital, Thai Red Cross Society, Bangkok, Thailand, for their invaluable contributions in implementing and facilitating the protocol outlined in this study.

## Author contributions

**Conceptualization:** Thananop Pothikamjorn, Thanapong Somkijrungroj, Marisa Prasanpanich, Nuntachai Surawatsatien, Wasee Tulvatana.

**Data curation:** Thananop Pothikamjorn, Nuntachai Surawatsatien.

**Formal analysis:** Marisa Prasanpanich.

**Funding acquisition:** Thananop Pothikamjorn, Thanapong Somkijrungroj, Wasee Tulvatana.

**Investigation:** Thanapong Somkijrungroj, Marisa Prasanpanich, Nuntachai Surawatsatien, Wasee Tulvatana.

**Methodology:** Thananop Pothikamjorn, Thanapong Somkijrungroj, Marisa Prasanpanich, Wasee Tulvatana.

**Project administration:** Thananop Pothikamjorn, Wasee Tulvatana.

**Writing – original draft:** Thananop Pothikamjorn, Thanapong Somkijrungroj, Marisa Prasanpanich, Nuntachai Surawatsatien, Wasee Tulvatana.

**Writing – review & editing:** Thananop Pothikamjorn, Thanapong Somkijrungroj, Marisa Prasanpanich, Nuntachai Surawatsatien, Wasee Tulvatana.

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
