## [Decision Letter · Decision Letter 0]

27 Jul 2025

Dear Dr. Tulvatana,

Thank you for submitting your manuscript to PLOS ONE. After careful consideration, we feel that it has merit but does not fully meet PLOS ONE’s publication criteria as it currently stands. Therefore, we invite you to submit a revised version of the manuscript that addresses the points raised during the review process.

We look forward to receiving your revised manuscript.

Kind regards,

Ogugua Ndubuisi Okonkwo, M.D.

Academic Editor

PLOS ONE

Journal Requirements:

Additional Editor Comments:

Dear Authors,

Apologies for the time taken to get back to you on your manuscript.

It took a bit of time securing the minimum 2 reviews required for your manuscript.

Reviewer 2 has made significant comments which if addressed will improve the quality of your protocol.

Please read through this in detail and respond to the comments.

Since this is a study protocol, results and discussions are not expected as suggested by reviewer one.

Thank you again for your patience during the review process.

Reviewers' comments:

Reviewer's Responses to Questions

**Comments to the Author**

1. Does the manuscript provide a valid rationale for the proposed study, with clearly identified and justified research questions?

Reviewer #1: Yes

Reviewer #2: Yes

2. Is the protocol technically sound and planned in a manner that will lead to a meaningful outcome and allow testing the stated hypotheses?

Reviewer #1: Yes

Reviewer #2: Yes

3. Is the methodology feasible and described in sufficient detail to allow the work to be replicable?

Reviewer #1: Yes

Reviewer #2: No

4. Have the authors described where all data underlying the findings will be made available when the study is complete?

Reviewer #1: Yes

Reviewer #2: Yes

5. Is the manuscript presented in an intelligible fashion and written in standard English?

Reviewer #1: Yes

Reviewer #2: Yes

You may also provide optional suggestions and comments to authors that they might find helpful in planning their study.

Reviewer #1: Thanks for invitation, Interesting title.

Methodology is quite perfect; however, surprsingely I was searching for the "Results"!!!

No result is mentioned and clarified. Too many tables were used, Table 1 and 2 and 4 are appropriate for appendix not the main text.

The authors too much enganged with the protocols (great however), they missed the main section: results and appropriate discussion!!!

Reviewer #2: In this manuscript, the authors present a detailed protocol for investigating ultrastructural and pathological characteristics of the internal limiting membrane (ILM) in cases with and without associated epiretinal membrane (ERM), using light and electron microscopy. The study addresses an important and understudied area and has the potential to provide valuable insights into ERM pathogenesis. However, several points require clarification or further elaboration to strengthen the protocol:

1. Please clearly define the indications for ILM peeling in the "ILM without ERM" group (e.g., macular hole,…). Additionally, clarify how the absence of ERM is confirmed clinically or histologically in these cases.

2. In some patients with ERM, ILM and ERM may not be separable during peeling. How are such cases handled? Will you apply double staining or exclude these cases from analysis?

3. The inclusion of patients with ERM due to silicone oil tamponade or prior intravitreal injections may introduce heterogeneity, as these etiologies may involve distinct pathological features. Please justify this inclusion or consider stratification.

4. There is a substantial imbalance in the expected number of specimens between groups (60–80 with ERM vs. 10–20 without ERM). Please explain the rationale behind this discrepancy and how it might affect statistical comparison.

5. In Table 2, the inclusion criterion “Participants who need ILM peeling” should be described in more detail, specifying surgical indications.

6. Also in Table 2, please clarify the exclusion criterion “Disease or surgical procedures which specimen harvesting method will complicate the surgery or the participant’s outcome” with specific examples.

7. The "Specimen Harvesting" section requires more detail. For example, will you perform double staining to differentiate ILM from ERM during combined membrane removal?

8. The number of operating surgeons and their level of experience should be specified, as this may impact the quality and consistency of specimen collection.

9. While the limitations of a cross-sectional design are acknowledged, it is worth emphasizing that this approach limits the ability to draw causal inferences regarding ERM development and progression.

10. Since all participants are recruited from a single tertiary center, the study population may not reflect broader demographics. Please address the potential for selection bias due to institutional surgical preferences or referral patterns.

Addressing these points will improve the clarity, reproducibility, and interpretability of the study protocol.

**Do you want your identity to be public for this peer review?** For information about this choice, including consent withdrawal, please see our Privacy Policy

Reviewer #1: **Yes: ** Masoud Mirghorbani

Reviewer #2: **Yes: ** Hamid Riazi-Esfahani

---

## [Author Response · Author response to Decision Letter 1]

7 Aug 2025

Editorial office comments

1.Please ensure that your manuscript meets PLOS ONE's style requirements, including those for file naming.

Response: Thank you for your valuable feedback. We have revised the manuscript and updated the file names to comply with PLOS ONE’s formatting and style requirements.

Response: Thank you for your comment. We have included the relevant grant numbers in the “Funding Information” section, as requested and ensure it matched the “Financial disclosure”.

Response: Thank you for the opportunity to clarify our data availability policy. All authors have agreed that all individual participant data collected during the trial, after de-identification, will be made available. The study protocol will also be accessible. Data will be available beginning 3 months and ending 5 years following publication of the full article, which is currently under review, to researchers who submit a methodologically sound proposal aimed at achieving the objectives outlined in the approved proposal. Proposals should be directed to waseetulvatana@chula.md. To gain access, data requestors will be required to sign a data access agreement. Data will be hosted for 5 years on a third-party platform (link to be provided upon publication). We have added this statement to the “Data Availability” section of the manuscript.

Response: Thank you for your suggestion. We have relocated all ethics-related content to the “Methods” section.

Response: Thank you for the clarification. The reviewers did not recommend any additional published works to be cited.

Reviewer’s #1 comments

1. Methodology is quite perfect; however, surprisingly I was searching for the "Results"!!!

No result is mentioned and clarified.

The authors too much engaged with the protocols (great however), they missed the main section: results and appropriate discussion!!!

Response: Thank you for your comments. As this is a protocol paper, results and the discussion of results are not applicable at this stage.

2. Too many tables were used, Table 1 and 2 and 4 are appropriate for appendix not the main text.

Response: Since no results have been generated, there are no results tables or figures included. We believe that all currently included tables, particularly Table 2 outlining the inclusion and exclusion criteria, are essential and appropriate for the manuscript.

Reviewer’s #2 comments

1. Please clearly define the indications for ILM peeling in the "ILM without ERM" group (e.g., macular hole,…). Additionally, clarify how the absence of ERM is confirmed clinically or histologically in these cases.

Response: Thank you for this valuable comment. We have revised Table 2 to include an example of surgical conditions warranting ILM peeling in the “ILM without ERM” group. Specifically, we added the following clarification: “Participants who require ILM peeling, such as for macular hole closure or ERM prophylaxis in those with RD.”

The absence of ERM will be confirmed using the current gold standard—OCT. In addition, we will document the surgical diagnosis at the time of specimen collection. In some cases, both ILM and ERM may be identified intraoperatively during brilliant blue G staining, particularly when OCT quality is suboptimal. These instances may occur in participants with vitreous hemorrhage, dense cataract, or severe RD. Such cases will be categorized in our analysis as “cannot evaluate” and will be appropriately stratified.

We have added these details into the “Statistical Analyses Plan” section as follows:

“The classification of ILM with or without ERM for analysis will be based on the current gold standard of diagnosis, OCT. Additionally, the surgical diagnosis at the time of specimen collection will be recorded, as both ILM and ERM may occasionally be identified intraoperatively using brilliant blue G staining, particularly in cases where preoperative OCT is of poor quality. Such scenarios may arise in participants with vitreous hemorrhage, dense cataract, or severe RD. These cases will be categorized as ‘cannot evaluate’ in our analysis and will be appropriately stratified.”

[Added in Page 22, Line 252 to 257]

2. In some patients with ERM, ILM and ERM may not be separable during peeling. How are such cases handled? Will you apply double staining or exclude these cases from analysis?

Response: Thank you again for your valuable comments. We aim to collect both ILM and ERM for the histopathological examination. In cases with separated ERM and ILM, these cases may need multiple tissue peeling within the same collection tube. In case with inseparable ILM and ERM, these specimen will be pulled out once or more as surgeon preference to have the largest amount of specimen available for the tissue processing.

To allow for separate histopathological evaluation, we note that ILM and ERM exhibit clearly distinct features in both light and electron microscopy, as previously reported by Smiddy et al., Gandorfer et al., Beyazyildiz et al., Kritzenberger et al., and Regoli et al. [Stated in table 3] We are confident that our pathologists are well-equipped to distinguish these characteristics.

We have revised the “Specimen harvesting” section of the manuscript accordingly, now stating:

“Subsequently, the surgeon will perform continuous curvilinear membrane peeling using forceps, targeting either the ILM alone or both the ILM and ERM, depending on intraoperative findings. In cases where the ILM and ERM are present together, we aim to collect both membranes for histopathological examination. If the ILM and ERM are separated, multiple peels may be required, and all tissue will be collected into the same specimen tube. In cases where the ILM and ERM are inseparable, the membranes may be removed in one or more passes at the surgeon’s discretion to obtain the maximum amount of tissue for processing. All specimens will be extracted from the intraocular space via the vitrectomy port using forceps.”

[Edited in Page 15, Line 186 to 194]

3. The inclusion of patients with ERM due to silicone oil tamponade or prior intravitreal injections may introduce heterogeneity, as these etiologies may involve distinct pathological features. Please justify this inclusion or consider stratification.

Response: Thank you for your valuable comments. We did include participants with silicone oil tamponade and patients with proliferative diabetic retinopathy (PDR), who had previously received intravitreal injections. These factors were incorporated in our secondary outcome analysis, stratifying idiopathic versus secondary ERM. Both silicone oil tamponade and PDR were considered causes of secondary ERM. We have updated Table 4 to provide detailed data on the causes of secondary ERM for stratification in our statistical analyses, as follows: “Cause of secondary ERM, including diabetes mellitus with NPDR or PDR, retinal detachment, retinal vascular occlusive diseases, uveitis, retinal breaks seen during surgery or from fundus photography, and vitreous status such as post-vitrectomy and silicone oil.”

4. There is a substantial imbalance in the expected number of specimens between groups (60–80 with ERM vs. 10–20 without ERM). Please explain the rationale behind this discrepancy and how it might affect statistical comparison.

Response: Thank you for your valuable comment. Our sample size estimation was based on the number of operations performed in the three months prior to IRB submission, extrapolated to a one-year specimen collection period. Statistical analyses will primarily use the Z-test for proportions [Stated in Table 1]. Although matched paired analysis (e.g., 1:1 matching) could strengthen statistical rigor, we opted not to perform this due to the limited number of specimens and budget constraints associated with additional specimen collection. To clarify this, we added the following statement in the “Sample Size” section: “We then extrapolated this to a one-year specimen collection period to estimate our sample size, although the actual number may vary depending on the surgical capacity and availability of both surgeons.” [Page 12, Line 158 to 160]

5. In Table 2, the inclusion criterion “Participants who need ILM peeling” should be described in more detail, specifying surgical indications.

Response: Thank you for this insightful comment. We have revised Table 2 to include an example of surgical conditions warranting ILM peeling in the “ILM without ERM” group. Specifically, we added the clarification: “Participants who require ILM peeling, such as for macular hole closure or ERM prophylaxis in those with retinal detachment.”

6. Also in Table 2, please clarify the exclusion criterion “Disease or surgical procedures which specimen harvesting method will complicate the surgery or the participant’s outcome” with specific examples.

Response: Thank you for your detailed comment. To further clarify our exclusion criteria, we have revised Table 2 to include: “Diseases or surgical conditions in which specimen harvesting would complicate the procedure or adversely affect the participant’s outcome, such as highly complex cases where tissue collection may hinder the surgeon’s ability to perform a safe surgery.”

7. The "Specimen Harvesting" section requires more detail. For example, will you perform double staining to differentiate ILM from ERM during combined membrane removal?

Response: Thank you for your comments. As previously mentioned, the ILM and ERM exhibit clearly distinct features in both light and electron microscopy, as previously reported by Smiddy et al., Gandorfer et al., Beyazyildiz et al., Kritzenberger et al., and Regoli et al. [Stated in table 3]

We have revised the “Specimen harvesting” section of the manuscript accordingly, now stating:

“Subsequently, the surgeon will perform continuous curvilinear membrane peeling using forceps, targeting either the ILM alone or both the ILM and ERM, depending on intraoperative findings. In cases where the ILM and ERM are present together, we aim to collect both membranes for histopathological examination. If the ILM and ERM are separated, multiple peels may be required, and all tissue will be collected into the same specimen tube. In cases where the ILM and ERM are inseparable, the membranes may be removed in one or more passes at the surgeon’s discretion to obtain the maximum amount of tissue for processing. All specimens will be extracted from the intraocular space via the vitrectomy port using forceps.”

[Edited in Page 15, Line 186 to 194]

8. The number of operating surgeons and their level of experience should be specified, as this may impact the quality and consistency of specimen collection.

Response: Two highly experienced retina surgeons (TS and NS), who are coauthors of this study protocol, will be the only surgeons performing the specimen harvesting. We stated: “All surgeries will be performed by two highly experienced retina surgeons (TS and NS), with participants recruited from their respective outpatient clinics.” This is detailed in the “Design and setting” section on page 11, Line 146 to 147.

9. While the limitations of a cross-sectional design are acknowledged, it is worth emphasizing that this approach limits the ability to draw causal inferences regarding ERM development and progression.

Response: Thank you for this insightful comment. We have added the following statement to the Discussion section on page 23 to 24, Line 284 to 289:

“Given the cross-sectional study design, our ability to draw causal inferences regarding ERM development and progression is limited. Additionally, this design does not allow for the assessment of ILM regeneration, ERM recurrence, or long-term visual outcomes. A longitudinal study would be required to correlate ultrastructural findings with disease progression and clinical outcomes over time.”

10. Since all participants are recruited from a single tertiary center, the study population may not reflect broader demographics. Please address the potential for selection bias due to institutional surgical preferences or referral patterns.

Response: Thank you for this interesting comment. We have included the following in the Discussion section on page 24, Line 289 to 292:

“Lastly, we acknowledge that our study population was recruited from a single tertiary care center, which may introduce selection bias due to institutional surgical preferences or referral patterns. This limitation may affect the generalizability of our findings to broader or more diverse populations and surgical settings.”

---

## [Decision Letter · Decision Letter 1]

13 Aug 2025

Comparative Electron Microscopy Analysis of Internal Limiting Membrane and Epiretinal Membrane Ultrastructure from Vitrectomy Surgery: A Study Protocol

PONE-D-24-37002R1

Dear Dr. Tulvatana,

We’re pleased to inform you that your manuscript has been judged scientifically suitable for publication and will be formally accepted for publication once it meets all outstanding technical requirements.

Kind regards,

Ogugua Ndubuisi Okonkwo, M.D.

Academic Editor

PLOS ONE

Additional Editor Comments (optional):

Reviewers' comments:

Reviewer's Responses to Questions

**Comments to the Author**

1. Does the manuscript provide a valid rationale for the proposed study, with clearly identified and justified research questions?

Reviewer #2: Yes

2. Is the protocol technically sound and planned in a manner that will lead to a meaningful outcome and allow testing the stated hypotheses?

Reviewer #2: Yes

3. Is the methodology feasible and described in sufficient detail to allow the work to be replicable?

Reviewer #2: Yes

4. Have the authors described where all data underlying the findings will be made available when the study is complete?

Reviewer #2: Yes

5. Is the manuscript presented in an intelligible fashion and written in standard English?

Reviewer #2: Yes

You may also provide optional suggestions and comments to authors that they might find helpful in planning their study.

Reviewer #2: All the comments are addressed properly and the revised manuscript is acceptable as a study protocol report.

**Do you want your identity to be public for this peer review?** For information about this choice, including consent withdrawal, please see our Privacy Policy

Reviewer #2: **Yes: ** Hamid Riazi-Esfahani

---

## [Editor Report · Acceptance letter]

PONE-D-24-37002R1

PLOS ONE

Dear Dr. Tulvatana,

I'm pleased to inform you that your manuscript has been deemed suitable for publication in PLOS ONE. Congratulations! Your manuscript is now being handed over to our production team.

Kind regards,

on behalf of

Prof Ogugua Ndubuisi Okonkwo

Academic Editor

PLOS ONE